# realSEUDO for real-time calcium imaging analysis

**Iuliia Dmitrieva**
Applied Math and Statistics
Johns Hopkins University
Baltimore, MD 21218, USA

**Sergey Babkin**
Microsoft Research
Redmond, WA, USA

**Adam S. Charles**
Biomedical Data Science
Center for Imaging Science
Mathematical Institute for Data Science
Kavli Neuroscience Discovery Institute
Johns Hopkins University
Baltimore, MD 21218, USA
adamsc@jhu.edu

## Abstract

Closed-loop neuroscience experimentation, where recorded neural activity is used to modify the experiment on-the-fly, is critical for deducing causal connections and optimizing experimental time. A critical step in creating a closed-loop experiment is real-time inference of neural activity from streaming recordings. One challenging modality for real-time processing is multi-photon calcium imaging (CI). CI enables the recording of activity in large populations of neurons however, often requires batch processing of the video data to extract single-neuron activity from the fluorescence videos. We use the recently proposed robust time-trace estimator—Sparse Emulation of Unused Dictionary Objects (SEUDO) algorithm—as a basis for a new on-line processing algorithm that simultaneously identifies neurons in the fluorescence video and infers their time traces in a way that is robust to as-yet unidentified neurons. To achieve real-time SEUDO (realSEUDO), we optimize the core estimator via both algorithmic improvements and an fast C-based implementation, and create a new cell finding loop to enable realSEUDO to also identify new cells. We demonstrate comparable performance to offline algorithms (e.g., CNMF), and improved performance over the current on-line approach (OnACID) at speeds of 120 Hz on average.

## 1 Introduction

Closed loop experiments enable neuroscientists to adapt presented stimuli or introduce perturbations (e.g., optogenetic stimulation) in real-time based on incoming observations of the neural activity. Such experiments are critical for both optimizing experimental time, e.g., by optimally selecting stimuli to fit neural response models [10], or by deducing causality by perturbing possible cause-and-effect hypotheses. Despite this critical need, closed loop experiments at the level of populations of single neurons is incredibly difficult as they require real-time processing of neural data, which can be computationally intensive to process. In particular, population-level recordings using modern technologies often require significant computation to extract individual neuronal activity traces, e.g., spike sorting for high-density electrode electrophysiology or cell detection in fluorescence microscopy [6, 9].

38th Conference on Neural Information Processing Systems (NeurIPS 2024).

One particularly challenging recording technology is fluorescence microscopy, in particular multi-photon calcium imaging (CI). CI has progressed significantly since its inception with optical advances enabling access to larger fields of view, and therefore higher data throughput. While neuroscientists now have access to hundreds-to-thousands of neurons at a time, the neuronal time traces embedded in the video as fluorescing objects. To extract each neuron's activity, multiple methods have been developed, including matrix factorization approaches, deep learning approaches, and others (we refer to a recent review for a more complete coverage of available methods and their nuances [6]).

Almost all current calcium image processing methods uses batch processing: i.e., using a full video all at once to identify the neurons in the data and their time traces. For example, a common approach is to identify cells in a mean image (the image containing the average fluorescence per pixel over all time) and then to extract the time-trace from the video given the neuron's location, e.g., by averaging pixels. Real-time processing does not afford such luxury. Instead, frames must be processed as they are collected. Furthermore minimal data can be stored and used, as large image batches reduce algorithmic speed. Finally, the incomplete knowledge of the full set of cells in the video can cause unintended cross-talk. Unidentified cells may overlap with known cells, causing a well-documented effect of false transients when the unknown cells fluoresce [12].

We thus present an algorithm capable of demixing CI data frame-by-frame in real-time. Our design goals are to operate at $> 30$ Hz with minimal temporary data storage (e.g., no buffering or initialization period needed) while minimizing false transient activity. Our primary contributions are: 1) An optimized SEUDO algorithm for fast, robust time-trace computation, 2) A new feedback loop to identify cells in real-time, and 3) patch-based parallelization that enables high-throughput calcium trace estimation across larger fields of view.

## 2  Background

Traditionally, CI analysis has been performed on full imaging videos. The goal of these algorithms is to extract from a pixel-by-time data matrix $\boldsymbol{Y} \in \mathbb{R}^{M \times T}$, where $M$ is the number of pixels in each frame and $T$ is the number of frames, a set of neural profiles $\boldsymbol{X} \in \mathbb{R}^{M \times N}$ (one for each of $N$ neurons) and a corresponding set of time traces $\boldsymbol{\Phi} \in \mathbb{R}^{N \times T}$. The former of these has, as each column of $\boldsymbol{X}$, a single component profile depicting which pixels constitute that fluorescing object, and how strong that pixel is fluorescing. The latter has as each row the corresponding time traces that represent how bright that object was at each frame. These time-traces are particularly important for relating neural activity to each other (i.e., modeling population dynamics) or to stimuli and behavior.

In typical approaches, full videos are required to either 1) identify summary images (e.g., mean or max images [24, 11, 25, 19]) to identify cells in, 2) to create a dataset within which points are clustered into cells [17, 31, 22, 27, 1, 28, 3, 28, 18], or 3) to perform simultaneous cell identification and demixing [26, 23, 8, 15, 16, 20, 21, 29, 13] (e.g., via matrix factorization or dictionary learning). For example, in the latter of these classes of algorithms, the data decomposition is solved via a regularized optimization, e.g.,

$$\widehat{\boldsymbol{X}}, \widehat{\boldsymbol{\Phi}} = \arg \min_{\boldsymbol{X}, \boldsymbol{\Phi}} \|\boldsymbol{Y} - \boldsymbol{X}\boldsymbol{\Phi}\|_F^2 + \mathcal{R}_X(\boldsymbol{X}) + \mathcal{R}_\Phi(\boldsymbol{\Phi}), \tag{1}$$

where $\|\cdot\|_F^2$ is the Frobenius norm (sum of squares of all matrix elements), and $\mathcal{R}_X(\boldsymbol{X})$ and $\mathcal{R}_\Phi(\boldsymbol{\Phi})$ are regularization terms for the profiles and time-traces, respectively. While many regularization combinations exist, common terms include sparsity in the neural firing, minimal overlaps, non-negativity, and spatial locality. Regardless, all methods require a large number of frames to identify the fluorescing components, with the exception of OnACID [14] and FIOLA [7].

OnACID and FIOLA operate in an on-line manner, utilizing the buffer of last $l_b$ residuals $\boldsymbol{r}_t = \boldsymbol{y}_t - \boldsymbol{X}\boldsymbol{c}_t - \boldsymbol{B}\boldsymbol{f}_t$ where $\boldsymbol{X}$ and $\boldsymbol{c}$ represent the spatial and temporal profiles of already recognized cells and $\boldsymbol{B}$ and $\boldsymbol{f}$ represent the spatial and temporal profiles of the known background signal. Both methods use a local Constrained Non-negative Matrix Factorization (CNMF) [26] in the spatial and temporal vicinity of that point. CNMF is an off-line algorithm that repeatedly performs alternating optimizations on $[\boldsymbol{X}, \boldsymbol{B}]$ and on $[\boldsymbol{c}, \boldsymbol{f}]$ using the full dataset, until it converges to a designated precision. Both methods require initialization periods, and FIOLA further requires GPU and CPU optimizaiton, raising the computational infrastructure costs. We seek a solution that does not need any initialization data and can be run on simpler CPU machines for easier incorporation into user's workflows.

## 2.1 Sparse Emulation of Unknown Dictionary Objects

One primary challenge in fully on-line settings is the incomplete knowledge of all fluorescing components at the experiment onset. Even in off-line methods, incomplete identification of cells can create scientifically impactful cross talk—termed *false transients*—in inferred activity [12, 16]. Another challenge is identifying new components from few frames: ideally from individual frames to reduce memory usage. Recent work has provided an algorithm with the potential to solve both challenges: The Sparse Emulation of Unused Dictionary Objects (SEUDO) algorithm [12].

SEUDO is a robust time-trace estimator for neuronal time traces. Given a single fluorescence video frame $y_t$, and a set of known profiles $X$, SEUDO models contamination from unknown profiles as $Wc$ where $W$ is a basis of small Gaussian bumps that linearly construct the interfering components, weighted by the sparse coefficients $c$ (i.e., most $c$ values are zero). SEUDO then solves the optimization

$$\widehat{\phi}_t = \arg \min_{\phi, c \geq 0} \left[ \min \left[ \|y_t - X\phi_t\|_2^2, \|y_t - X\phi_t - Wc\|_2^2 + \lambda\|c\|_1 + \gamma \right] \right], \qquad (2)$$

where $\lambda$ and $\gamma$ are model parameters and the internal $\min$ selects from the two internal expression that which has the minimal value. Since SEUDO operates per-frame, $y_t$, $Phi$, $c$ are all vectors. While SEUDO has demonstrated the ability to remove false transients [12], SEUDO's application has been limited to off-line post-processing due to: (1) slow computational speed, and (2) the need for pre-defined profiles $X$.

## 2.2 The FISTA Algorithm

The computational bottleneck in SEUDO is a weighted LASSO [32] optimization, which can be implemented with the Fast Iterative Shrinkage-Thresholding Algorithm (FISTA), which implements a momentum gradient descent [4]. FISTA optimizes

$$\min \left[ F(x) \equiv f(x) + g(x) : x \in \mathbb{R}^n \right], \qquad (3)$$

where $f(x)$ is a smooth convex function with Lipschitz constant $L > 0$, such that $\|\nabla f(x) - \nabla f(y)\| \leq L\|x - y\| \forall x, y \in \mathbb{R}^n$, e.g. in SEUDO $f(x) = \|y_t - X\phi_t - Wc\|_2^2$, and $g(x)$ is a continuous convex function that is typically non-smooth, e.g., $g(x) = \lambda\|x\|_1$. Each descent step of FISTA consists of an ISTA descent step and a momentum step:

$$x_k = \arg \min_x \left[ g(x) + \frac{L}{2} \left\| x - \left( y_k - \frac{1}{L}\nabla f(y_k) \right) \right\|^2 \right] \qquad (4)$$

$$y_{k+1} = x_k + \eta_k(x_k - x_{k-1}), \qquad (5)$$

where the parameter $\eta_k$ gradually reduces with $\eta_k = \frac{t_k - 1}{t_{k+1}}$, $t_{k+1} = \frac{1 + \sqrt{1 + 4t_k^2}}{2}$, $t_1 = 1$.

## 3 Real-time SEUDO

Here we develop Real-time SEUDO (realSEUDO) that resolves the primary limitations of SEUDO and extends the algorithm significantly from a time-trace estimator to a real-time cell identification method. Specifically we improve the computationally intensive momentum descent algorithm used to solve Equation (2) by reducing the number of steps of momentum descent, implementing parallelism, optimizing internal computations (e.g., of smoothness parameters), and reducing the complexity of the original fitting problem without a substantial loss of quality by manipulating its inputs. Moreover we add a new algorithm that automatically recognizes the neurons that have not previously been seen and adds them to the dictionary of known components. Finally, we implement our framework with a patch-based parallelism that avoids the computational scaling of LASSO in higher dimensions.

At a high level, the realSEUDO algorithm (Alg. 1, Fig. 1) operates as follows: realSEUDO is initialized with zero known components (an empty set). When fluorescence activity in a frame reaches threshold, an event is triggered that saves the activity profile of that event as a temporary candidate profile. Profiles are moved from the temporary profiles to the static set of profiles if they remain active for a sufficient number of frames . The static profile set is then used to identify the activity of those components in future frames, with unexplained components becoming candidate profiles and cycling back into the temporary profiles, followed by an update of the static profile set.

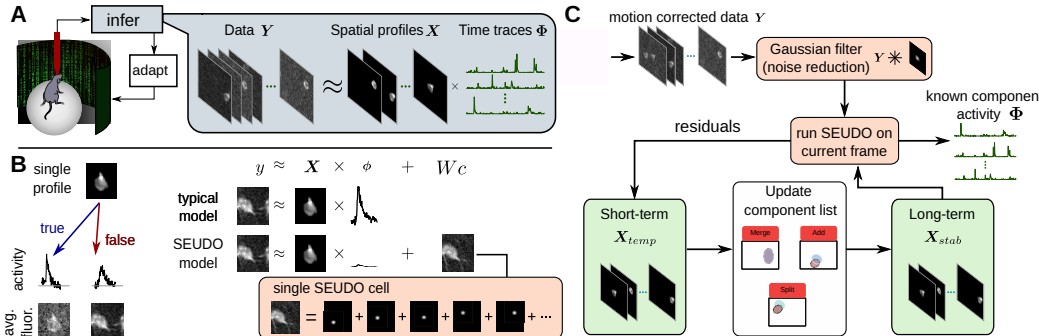

Figure 1: The realSEUDO algorithm. A: Real-time inference of cells and their activity from calcium imaging is crucial to closed-loop experiments, however, Typical CI demixing requires batch processing, e.g., via matrix factorization. B: realSEUDO builds on the robust SEUDO algorithm that prevents activity in missing or unknown cells from creating false activity in known cells by explicitly modeling contamination as a sparse sum of small Gaussian blobs (right). The sum of the estimated Guassians further provides an approximation of shape of the unknown cells, which can be used to seen new known cells. C: We propose a method based around the SEUDO estimation algorithm that can identify cells in real time by robustly removing known cells and using the residuals to identify new cells in the data.

## 3.1 SEUDO optimization

The first requirement of realSEUDO is a fast implementation of SEUDO that can operate at $> 30$ fps. We achieved this requirement through a combination of efficient implementations, algorithmic optimization, and updates to the base SEUDO model.

**C++ implementation:** The original SEUDO implementation used the TFOCS MATLAB library [5]. We thus first improved SEUDO's run-time by switching from MATLAB's interpreted programming language running TFOCS to a fast implementation of LASSO [32] via FISTA [4] in the C++ compiled language. To further improve performance, we optimized the C++ code with the use of templates to eliminate the function call overhead in tight loops, and also employ parallelism, based on the POSIX threads with TPOPP library wrapper [2]. To prevent a bottleneck from the passing of data through Matlab's OOP API at the MATLAB/C++ interface, we switched to the non-OOP version of the MATLAB-to-C API.

While beneficial, the C++ SEUDO implementation did not alone achieve the desired processing rate. We further improved runtimes by optimizing the cost function and derivative computations. The partial LASSO component of SEUDO that performs optimization at each frame $y$ using FISTA can be written as $\arg\min f(\psi) + \lambda g(\psi)$, where $f = \|y - \chi\psi\|_2^2$ is the least-squares term and $g = \|\psi\|_1$ is the $\ell_1$ penalty. In FISTA, the profile time traces and Gaussian kernels are unified in one vector, i.e., $\psi$ is a concatenation of $\Phi$ and $c$, and $\chi = [X, W]$. With $M$ as the number of pixels per frame, $N$ as the number of neurons, and $K$ as the number of Gaussian kernels, the set of problem dimensions are $y \in \mathbb{R}^M, \chi \in \mathbb{R}^{M \times (N+K)}, \psi \in \mathbb{R}^{N+K}, \lambda \in \mathbb{R}^{N+K}$, with the first $N$ elements of $\lambda$ corresponding to $\Phi$ being equal to 0.

In FISTA, a number of internal computations become bottlenecks; in particular computing the gradients $\nabla_\psi f$ and $\nabla_\psi (f + g)$, the Lipshitz smoothness estimation, and the momentum/stopping criteria.

**Gradient computation:** We reduce the burden of the gradient computations by both reducing the number of times the gradient must be used, and by improving the internal gradient computation. For the former, we note that naive implementations compute both a step in the direction of $\nabla_\psi f$ and then in the direction of $\nabla_\psi (f + \lambda g)$. Moreover, we note that these two steps in slightly different directions cause the gradient to dither around the optimum. We thus instead only take a step in the direction of $\nabla_\psi (f + \lambda g)$ (similar to [4]). For the latter, computing $\nabla_\psi (f + \lambda g)$ requires matrix vector multiplications with $\chi$ and its transpose. Since $\chi$ is sparse, we save memory and computation, by generating $\chi$ on-the-fly via convolutions instead of storing it in memory. Specifically, the gradient

$\nabla_{\boldsymbol{\psi}} f$ requires computing $\boldsymbol{\chi}^T \boldsymbol{\chi} \boldsymbol{\psi}$, which we reorganize to compute in two passes:

$$\boldsymbol{v}_j = \sum_{1 \leq i \leq \boldsymbol{N}+\boldsymbol{K}} \boldsymbol{y}_j - \boldsymbol{\chi}_{ji}\boldsymbol{\psi}_i, \qquad \frac{\mathrm{d}f}{\mathrm{d}\boldsymbol{\psi}_m} = 2 \sum_{1 \leq j \leq \boldsymbol{M}} \boldsymbol{\chi}_{jm}\boldsymbol{v}_j. \tag{6}$$

The first pass (Eqn. 6, left) computes a set of intermediary variables, and the second pass (Eqn. 6, right) uses these values to compute the gradient dimensions. The two pass approach factors out repeated computations, reducing the complexity from $O(n^3)$ to $O(n^2)$. Moreover, the computation of each pass is highly parallelizable by partitioning of the first pass by $j$, the second by $m$, and efficiently skipping the iterations over the zero elements in the sparse matrix $\boldsymbol{\chi}$.

**Lipshitz constant:** To improve the efficiency of estimating the Lipschitz constant $L$, note that $L = \max(\|\nabla f(\boldsymbol{x}_1) - \nabla f(\boldsymbol{x}_2)\|/\|\boldsymbol{x}_1 - \boldsymbol{x}_2\|)$. For our cost function, we can approximate $L$ with independent computations in each dimension. This estimation reduced the number of steps by as much as 30% over typical computations of $L$ before each step based on the local gradient.

**Momentum:** The momentum descent central to solving the partial LASSO tends to spend many steps on stopping the momentum, especially with the large values of the Lipschitz constant $L$ (i.e. the non-momentum steps are small). One such case is "circling the drain" around the minimum, with the momentum causing the overshoot in one dimension while another dimension is stopping. Another case is when a dimension is moved past the boundary (e.g., $\boldsymbol{x} \geq 0$ for SEUDO), where the solution pushes past the boundary into negative values. This produces suboptimal solutions and increasing the number of steps necessary. FISTA includes a parameter $\eta$ that progressively limits the top speed of descent to reduce such problems. We improved these cases by resetting the momentum to zero on a dimension when it either attempts to cross into the negative values or when its gradient changes sign. The dimensional momentum stopping stops abruptly at the right time, obviating the need for slowing and thus we can simplify FISTA by fixing $\eta = 1$.

Our momentum stopping can further extend to broader optimization problems. We demonstrated this ability on momentum optimization in neural network training (see Supplement), where it provided a substantial improvement. Our modified FISTA produced the same error rate and squared mean error as gradient descent in about 10 times fewer training passes, or about 10 times lower error rate and 1.5 times lower mean square error in the same number of passes. The full modified FISTA algorithm is presented in Algorithm 2 (see Supplement).

**SEUDO model adjustments:** As a third step, we modified the SEUDO optimization program to achieve the final speedups. The original SEUDO spaced the Gaussian components in $\boldsymbol{W}$ by one pixel, which we found to be highly redundant. The kernels with radius $r$ cover $(\pi * r^2)$ pixels, and thus each pixel is covered by $(\pi * r^2)$ kernels. This redundancy results in FISTA continuing to adjust the kernel coefficients $\boldsymbol{c}$ after the neural activations $\boldsymbol{x}$ converge. Reducing the number of kernels thus reduces both the number of gradient descent steps and the per-step cost, accelerating the computation more than quadratically. For a kernel with diameter of 30 pixels this improves the performance by a factor of over 100 without substantial degradation of false transients removal or the recognition of the interfering components' shape $\boldsymbol{W}\boldsymbol{c}$.

We benchmarked our speedups against the original SEUDO on 45000 frames across 50 cells from [12]. SEUDO ran at 5.8-6.9 s/cell on a Macintosh M1. The optimized C++ implementation without the MATLAB C++ API reduced the runtime to 0.9-1.1 s/cell (a 6-7x improvement). Sparse SEUDO provided further acceleration to a run time of 0.2 s, with 0.1 s for the computation and 0.1 s for the overhead of converting data between Matlab and native code; a total of a 29-34.5x speed-up, allowing SEUDO to run in real time.

## 3.2 Automatic cell recognition

We next developed the cell recognition feedback loop that completes the realSEUDO algorithm. To minimize data storage and compute, we designed realSEUDO to run on a frame-by-frame basis. At a high level, our automatic cell recognition first runs SEUDO on the current incoming (denoised) frame given the currently identified profiles $\boldsymbol{X}_{stab}$. The loop then identifies contiguous bright areas in the residual frames, i.e., the SEUDO cells $\boldsymbol{W}\boldsymbol{c}$, and places them in a 'temporary profile' array $\boldsymbol{X}_{temp}$. The temporary profiles are then updated (via merging with new potential profiles) given new, incoming, frames until they are stable and moved to the stable, known profile list $\boldsymbol{X}_{stab}$ that is updated less frequently by addition, merging and splitting of temporary profiles.

Procedurally, we first preprocess each incoming frame to reduce noise and improve profile recognition. Calcium imaging analysis often uses running averages in space and time for noise reduction. We thus implement both a spatial Gaussian filter, as well as a running average of several sequential frames. We keep the window length as a tunable parameter that can be set to one for frame-by-frame processing with minimize temporal blurring.

We estimate the activation level in the denoised frame for each of the stable profiles $X_{stab}$ using the our fast SEUDO implementation. SEUDO returns the activation level $\phi_{kt}$ for the $k^{th}$ profile at time $t$ along with a robust residual that contains the structured fluorescence not captured by $X_{stab}$. We then run the residual through SEUDO a second time using the temporary profiles $X_{temp}$ to test if any temporary profile matches the frame's fluorescence and should be moved from $X_{temp}$ to $X_{stab}$. The residual after the second SEUDO application represents completely unknown profiles and are analyzed separately to determine if a new member of $X_{temp}$ should be created.

The detection of new temporary profiles is based on finding the areas of the image above the noise level. The noise level is evaluated by noting that most of each video frame has no activity, indicating that the median pixel value will be very close to the median value of the pixels in an all-dark frame containing the same noise. The half-amplitude of the noise $\sigma_{1/2}$ can be estimated as:

$$\sigma_{1/2} = \text{median}(\boldsymbol{y}_t) - \min(\boldsymbol{y}_t). \tag{7}$$

In some cases different areas of the image may contain a different amount of background lighting (e.g., changes in neuropil), which can skew the noise estimate. To overcome this challenge, we split the larger image into sections, with each section computing a local median which is smoothly interpolated between section of the image. This can be thought as either a krigging procedure or a cheap approximation to a local median evaluated independently for each pixel in the image.

To merge new profiles into the exiting profiles when adding new profiles to $X_{temp}$, we compute an overlap score. The scores aims to capture the following logic: If two temporal profiles are a close match in cross-section, they likely represent the same cell and should be merged. If they overlap only partially, they likely represent separate cells. If one cross-section is inside the other, look at relative brightness: if the smaller cross-section is also weaker, it's likely a weaker partial activation of the same cell and should be merged, if the smaller cross-section has a close or higher brightness, the larger cross-section likely represents an intertwining of two cells that has to be split.

The overlap computation thus is not a plain spatial overlap but includes a heuristics that identifies when one profile is mostly contained in another. The condition for merging two profiles in $X_{temp}$ is based on the comparison of numbers of common and unique pixels between profiles, where $P_1$ and $P_2$ are numbers of pixels in each of two profiles, $U_1$ and $U_2$ are the numbers of unique pixels in each profile, $C$ is the number of common pixels, $B_1$ and $B_2$ are the perimeters of bounding boxes for each profile, and $k_{temp}$ is a constant with an empirically chosen value of 0.75:

$$U_1 \leq B_1 * 0.5 \quad \text{or} \quad U_2 \leq B_2 * 0.5 \quad \text{or} \quad C \geq k_{temp} * \min(P_1, P_2) \tag{8}$$

After a profile is moved from $X_{temp}$ to $X_{stab}$, a different score is computed pair-wise between the new profile and each existing profile, to decide whether they should be left separate, or merged, or one of profiles split. If a merge or split is performed, the original profiles are removed from $X_{stab}$ and the results entered recursively into $X_{stab}$ as new profiles. The score for two profiles $A$ and $B$ in $X_{stab}$ is computed based on the brightness and measures of least-squares fit of the cells into each other 1) as whole cells (i.e., $\alpha_{AB} = \langle A, B \rangle / \langle A, A \rangle$) and 2) using only the overlapping region (i.e., $\beta_{AB} = \langle A_{ol}, B \rangle / \langle A_{ol}, A_{ol} \rangle$ where $A_{ol}$ is the profile $A$ restricted to the region overlapping with $B$). $\beta_{AB}$ is used as a measure of difference in brightness, against which the fit of the whole cells $\alpha_{AB}$ is compared as a measure of proximity in shape. Specifically we compute two ratios $\rho_{AB}$ and $\rho_{BA}$ are computed as

$$\rho_{AB} = \frac{\alpha_{AB}}{\beta_{AB}}, \qquad \rho_{BA} = \frac{\alpha_{BA}}{\beta_{BA}}. \tag{9}$$

A higher value of $\rho_{AB}$ (which is always $\leq 1$) means that cell A fits better inside cell B. The value of 1 means that it fits entirely inside cell B. The same principle applies symmetrically to $\rho_{BA}$.

### 3.3 Patching and profile matching

realSEUDO, although highly efficient for smaller patches, is still based on the LASSO algorithm that reduces in efficiency with much larger frames. Thus we adopt a patching scheme that breaks each

frame into small patches that can be parallelized to maintain the high framerate by utilizing the multi-threading in many modern processors. Patching, however, requires matching profiles across patches. Traditionally profiles discovered in data split into patches is to add overlap margins to the patches and to use profiles overlap in this region to determine matchings in neighboring patches. Additional margins, however, introduces redundant computation and decreases computational efficiency.

In realSEUDO we note that the logic behind the scoring we use to merge profiles in $X_{stab}$ and $X_{temp}$ within each patch can also be used to score the match of profiles across patches. Specifically, we extend the matching to include the profile temporal activity as an additional dimension to find matching cells in neighboring patches via consecutive gluing of the profiles. The highest score is assigned to the bidirectional match of both spatial and temporal dimensions, a lower score to symmetrical match of spatial dimensions and asymmetrical match of temporal dimension, a yet lower score to an asymmetrical match of both kinds of dimensions. We have observed successful matches even with zero margin, using the neighboring strips of pixels around the perimeters of the patches as the spatial dimension for matching.

## 4 Results

**Validation metrics:** To validate our approach we note that the main goal of realSEUDO, as with most functional imaging analyses, primarily aims to recover the time-traces of neural activity as accurately as possible [6, 12]. This means while the general location of neurons is important, metrics such as the the Intersection over Union (IoU) are too strong; i.e., the full set of pixels identified is not necessarily the important quantity. We instead compute the "unique neurons found". This metric aims to capture the need to know that the time-traces 1) correspond to real neurons in the data and 2) accurately reflect the temporal activity that will be used to study neural activity with respect to stimuli and behavior. The Unique Neurons Found (as defined in [30]) requires both that ROIs well align with known ROIs spatially (overlap of at least 50% of pixels) and that the time-trace correlation exceeds 0.5.

**Simulated data experiments:** We first applied all three algorithms to a simulated video created with Neural Anatomy and Optical Microscopy simulation (NAOMi) [30]. Specifically we simulated the neural activity over 20000 frames at 30Hz with fame size of 500x500 pixels. There are approximately 450 cells visible in this dataset (i.e. fluorescing cells intersecting the plane of imaging). We benchmarked the patch-based parallel processing of realSEUDO 80x80 pixel patches. For comparison we ran the off-line CNMF (a staple batch-based calcium imaging demixing algorithm) and OnACID, the computationally similar on-line method. realSEUDO found 201 true cells, identified as strongly correlated with ground-truth cells, while OnACID found 152 and CNMF (the offline method) found 308 (Fig. 2A-B). Furthermore, both realSEUDO and OnACID found many fewer false positives than CNMF, presumably because they cannot be fooled by small fluctuations integrated over the full recording (Fig. 2C-D). Note that to remove the confound of post-processing we followed prior work [30] in using the CNMF raw fluorescence traces instead of the model-based denoised traces. On average, realSEUDO processed 67.8 frames per second end-to-end, while OnACID ran at 9.2 fps.

**Applications to *in-vivo* mouse CA1 recordings** We applied realSEUDO to an *in vivo* calcium imaging recordings from mouse hippocampal area CA1 previously described in Gauthier et al. 2022 [12]. They consisted of 36 videos, each sized 90x90 pixels with 41750 frames sampled at 30 Hz. The outputs had previously been verified manually by Gauthier et al. 2022 [12] with human labeling of CNMF outputs. We applied realSEUDO to all videos and compared the outputs with the current online cell demixing algorithm OnACID [14], as well as a popular offline algorithm, CNMF [13], as an additional baseline.

We benchmarked realSEUDO against OnACID (an online analysis tool) and CNMF (an offline analysis tool) on real in-vivo calcium imaging movies (Fig. 3). Algorithmic performance was measured on an x86-64 computer with 48 CPU cores (Intel Xeon 6248R), 2 hyperthreads per core, 78 GB of memory, and without the use of a GPU. The initialization times were not included. On average, realSEUDO processed 162 frames per second compared to 26 processed by OnACID and 13 by CNMF: an improvement of 6.5x and 12.5x respectively (Fig. 3C). Quality-wise, we found that OnACID exhibited difficulties with adapting to larger ranges of pixel brightness, sometimes missing bright cells. Scaling pixel values improved OnACID results, but only mildly (one additional cell). Numerically OnACID and CNMF appear similar but they identified different components. SEUDO

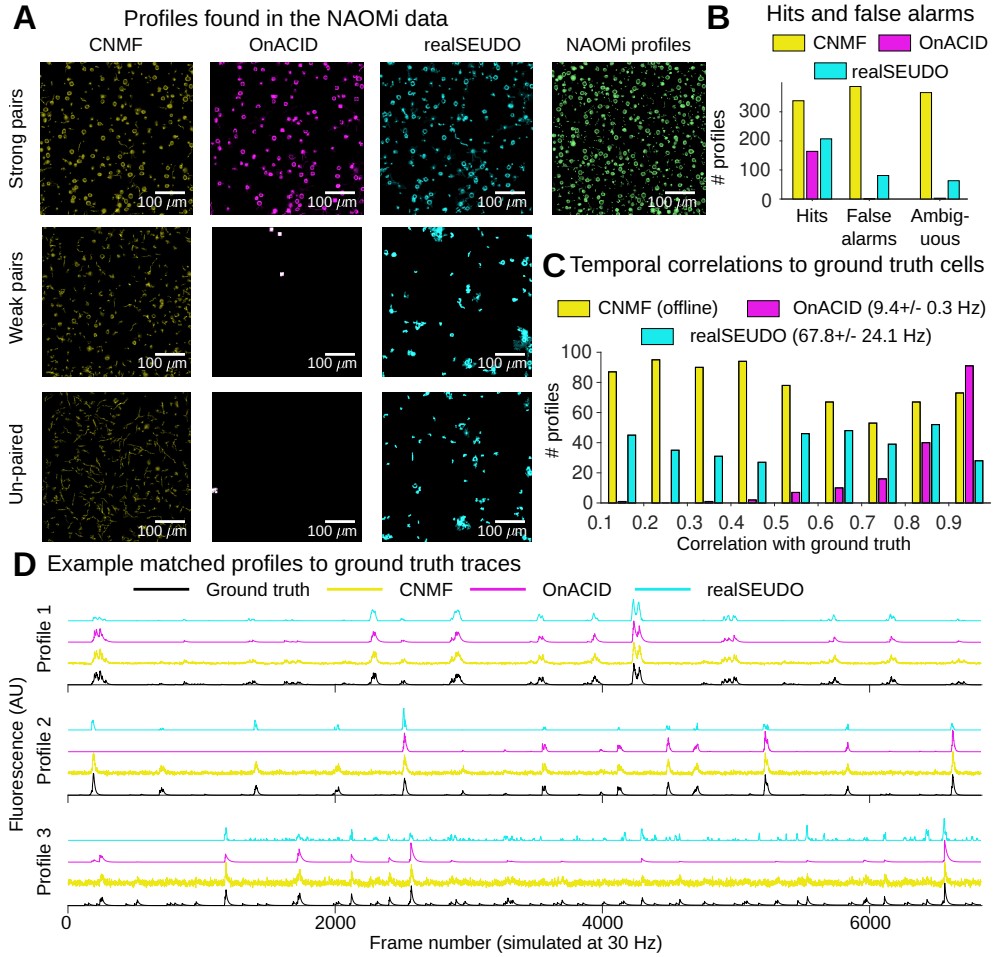

Figure 2: NAOMi results: A) Found cells in NAOMi for CNMF, OnACID and realSEUDO separated into Hits (strong or weakly correlated) and false alarms (uncorrelated). B) realSEUDO finds more cells than OnACID with minimal false positives. C) Temporal correlations for found "hits". D) Examples time-traces show correlation to ground truth.

results were most similar to CNMF, and with additional cells identified, and less false positives (Fig. 3B). Finally, the per-transient manual classification provided by [12] enabled us to assess if realSEUDO inherited the false transient removal properties of SEUDO. For a reasonable value of $\lambda = 0.15$, realSEUDO had a true positive rate of 75% and a false positive rate of 24%. While these numbers are a bit lower than the numbers reported in [12], in that study the authors average N=3 frames to reduce noise, while maintained single-frame analysis. This can be evident by the fact that missed transients were very small: realSEUDO kept 98% of real fluorescence and only 15% of false fluorescence.

**Additional *in-vivo* tests:** As final test we applied all three algorithms (realSEUDO, OnACID, CNMF) to a 2000-frame mesoscope video example collected by the Yuste lab at Columbia University and provided with the OnACID github package as a demo. For this example we similarly saw improved cell detection and runtime improvement in terms of fps (Fig. 3B).

## 5 Discussion

We present here an online method for cell detection and fluorescence time-trace estimation from streaming CI data: realSEUDO. realSEUDO is based on the SEUDO robust time-trace estimator that reduces bias due to unknown cells while also providing approximate shapes of the unknown fluoresc-

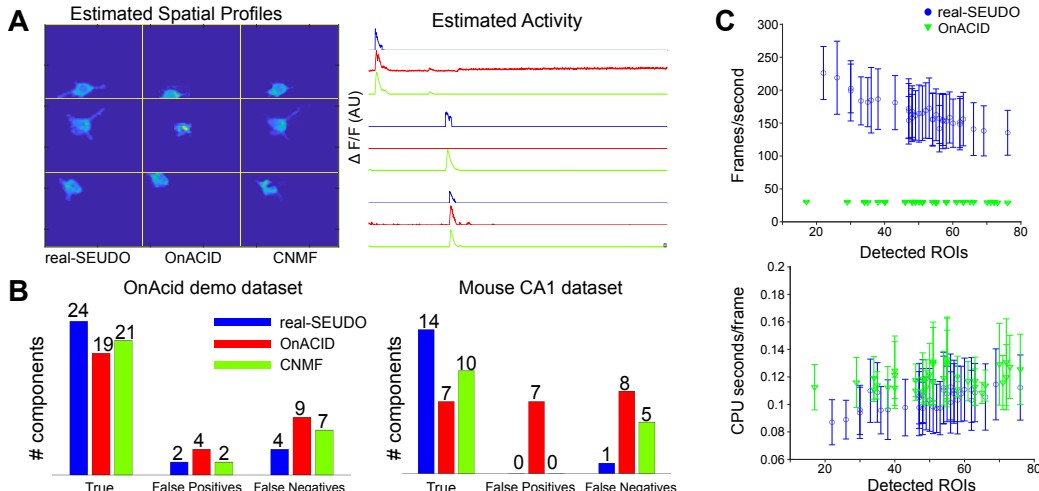

Figure 3: (A) Selected cell profiles and time-traces generated by realSEUDO, OnACID, and CNMF respectively on a subset of 2000 frames from a single image patch. (B) Counts of true positive, false positive and false negative cells found by each algorithm for two different recordings: all 41,750 frames data from one video from Gauthier et al. [12] (right) and from the OnACID demo (left). (C) Top: Total computational performance as a function of the number of detected cells for realSEUDO and OnACID, evaluated on the full set of 36 movies from Mouse CA1. Bottom: CPU use in CPU seconds per frame as a function of the number of detected cells for realSEUDO and OnACID, evaluated on the same 36 recordings.

ing objects. To build realSEUDO we 1) improved SEUDO's runtime via significant modifications at the code, algorithmic, and model levels 2) built a new feedback loop that allowed SEUDO (that has no cell finding component currently) to identify cells in real-time. Overall, realSEUDO can achieve frame processing rates of 80-200 fps, depending on cell density. While our goal was to exceed the typical 30 Hz data collection rate common to many experiments, the high processing efficiency leaves additional time to compute feedback in future closed loop systems. Moreover, realSEUDO can scale with faster recording rates as calcium indicators become faster, e.g., GCaMP8 [33].

realSEUDO's implementation exhibits a higher degree of parallelism than OnACID, however both are likely constrained by the employed tools. Specifically, the measurements in Fig. 3C show very little fps fluctuation with respect to cell count for OnACID, which can be due to a bottleneck in a single thread. realSEUDO has a lower latency to the first events for a new cell, OnACID requires a history of 100 frames to recognize a cell, while realSEUDO would produce the first events starting with the first frame that passes the low brightness threshold. Alternatively, OnACID has a higher native scalability with respect to the frame area, conditioned on similar cell counts. This likely result from OnACID's algorithm restriction of processing to areas in the immediate vicinity of known cells. realSEUDO can still achieve a high processing efficiency with reasonable hardware with our parallelization. Future work may further improve realSEUDO runtime by blanking out entire patches until activity is detected via simpler detectors.

One strength of realSEUDO is that it can be initialized with either an empty profile set. This ability to start from nothing will be useful in mesoscope settings when the field-of-view can be changed on-the-fly. Not requiring an initialization step will reduce start-up overhead at new fields of view. Furthermore, many of our speed adaptions deviate from the traditional gradient descent approach. In particular the Lipshitz constant approximation and the changes in the momentum and stopping criteria. In other domains, in particular for training deep neural networks, these deviations may also provide significant speedups, the extent of which should be quantified across broader applications in future work.

**Limitations:** While our results achieve the design criteria we initially set out, there are some potential barriers. For one, as with many real-time systems, the compute environment is very important to configure correctly. We have found the importance of explicitly setting the Linux CPU manager to enable the performance mode. The default automatic adaptable mode does not react properly to CPU loads of less than 100% of the whole capacity, and significantly skews the benchmarking by running

the CPUs at low frequency. These challenges are unfortunately necessary to achieve high levels of throughput without specialized hardware. Future work should develop walkthroughs and automated tools to guide the installation of the tools.

While our core algorithm is written completely in C++, and thus open source, we have found MATLAB convenient and efficient as a wrapper for prototyping wrappers for our core functions. Further work will add Python wrappers to allow for seamless integration into both MATLAB and Python pipelines, enabling realSEUDO to be more widely used.

Finally, we focused here only on cell detection, assuming access to the on-line motion correction algorithm from OnAcid. This focus may require additional packages to be handled by users for motion correction. We will further aim in future iterations to extend the core package to include motion correction and delta-F over F computations in order to reduce communication overhead and ease adoption by users. Moreover, to fully optimize the package for speed, these steps should be more holistically incorporated in C++ as well.

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

# A   Appendix

---

**Algorithm 1** realSEUDO Algorithm

---

1:  Initialize: $\boldsymbol{X}_{temp} \leftarrow []$; $\boldsymbol{X}_{stab} \leftarrow []$
2:  **for** each frame $\boldsymbol{y}_t$ **do**
3:      Denoise $\boldsymbol{y}_t$
4:      Identify the profiles in the current frame
5:      **for** all new profiles **do**
6:          **if** current profile overlaps any of $\boldsymbol{X}_{temp}$ **then**
7:              Merge new $\boldsymbol{X}_{temp}$ profiles into the current $\boldsymbol{X}_{temp}$
8:          **else**
9:              Add the current profile to $\boldsymbol{X}_{temp}$
10:         **end if**
11:     **end for**
12:     **for** all profiles in $\boldsymbol{X}_{temp}$ that have not been updated in the last few frames **do**
13:         Move them from $\boldsymbol{X}_{temp}$ to $\boldsymbol{X}_{stab}$
14:         Merge the moved profiles with existing $\boldsymbol{X}_{stab}$ profiles
15:     **end for**
16:     $\phi_t, \boldsymbol{r}_t \leftarrow \text{SEUDO}(\boldsymbol{y}_t, \boldsymbol{X}_{stab})$
17:     Report $\phi_t$ as detections
18:     $\phi_t' \leftarrow \text{SEUDO}(\boldsymbol{r}_t, \boldsymbol{X}_{temp})$
19:     **for** each profile $k$ in $\boldsymbol{X}_{temp}$ **do**
20:         **if** the $\phi_{kt}' > \gamma$ or this profile was previously active **then**
21:             Report this activation as early detection
22:         **end if**
23:     **end for**
24: **end for**

---

## A.1   Application of modified FISTA to neural network optimization

We further tested the modified FISTA momentum descent algorithm to problems outside of neuroscience—the training of neural networks—to evaluate the scope of applicability of our improvements.   We used the problem of recognition of handwritten digits on a data set from AT&T Research available at `https://hastie.su.domains/StatLearnSparsity_files/DATA/zipcode.html`, reduced to 8x8 pixels, with a training set of 7291 images. The neural network (NN) model used the Leaky ReLU activation, with layer sizes 64, 64, 32, 10.

The common approach to training NNs uses stochastic gradient descent, including stochastic momentum methods. Thus to compare to a non-stochastic momentum method we established a non-stochastic baseline.

The dynamic estimation of the Lipshitz constant $L$ from TFOCS cannot be applied to the neural network optimization because the optimization cost is highly non-linear.  The multi-dimensional estimation of $L$ is also computed only for the specific SEUDO function. The common practice is to use a fixed descent rate, which serves as an analog of $\frac{1}{2L}$. Estimating the highest descent rate is still not a fully solved problem. Algorithms for dynamic evaluation of the descent rate do exist (e.g., the ADAM algorithm in Kingma & Ba, 2015), however they rely on a constant to be picked for a particular problem.

The advantage of stochastic methods is that they can use a higher descent rate without diverging, as seen by observing the dependency of logarithm of mean square error from the number of training passes for various descent rates (Fig 6). In these results we ensure that we start from the same fixed randomized initial state since different initial states can produce wildly different results.

We observe that for the same descent rate (0.05 per pass), the stochastic and non-stochastic methods produce very similar error values, however the graph for the stochastic method is more smooth. The roughness of the graph represents the small divergences that manage to converge again over time, and shows that the descent rate is close to the maximum. However the stochastic method can accommodate a 100 times higher descent rate without diverging, and even a 1000 higher descent

**Algorithm 2** Modified FISTA algorithm
---
1: Initialize $t = 1$; $x[] = (initial\ values)$; diff$[] = [0]$; gradient_last$[] = [0]$
2: **for** $step = 1$ to $maxstep$ **do**
3:     $t_{next} = \frac{1+\sqrt{1+t^2*4}}{2}$
4:     $\eta = \frac{t_{-1}}{t_{next}}$
5:     **if** $step \neq 1$ **then**
6:         $t = t_{next}$
7:     **end if**
8:     $x[] = x[] + \eta * \text{diff}[]$
9:     **for** each $i$ in dimensions of $x$ **do**
10:         **if** $x[i] < 0$ **then**
11:             $x[i] = 0$; diff$[i] = 0$;
12:         **end if**
13:     **end for**
14:     gradient$[] = \text{compute\_gradient\_f}(x[])$
15:     $x[] = x[] - \frac{gradient[]}{L}$
16:     **for** each $i$ in dimensions of $x$ **do**
17:         **if** $x[i] < 0$ **then**
18:             $x[i] = 0$; diff$[i] = 0$; gradient$[i] = 0$
19:         **else**
20:             **if** gradient$[i] *$ gradient_last$[i] < 0$ **then**
21:                 diff$[i] = 0$
22:             **else**
23:                 diff$[i] = \text{diff}[i] - \frac{\text{gradient}[i]}{L}$
24:             **end if**
25:         **end if**
26:     **end for**
27:     gradient_last$[] = $ gradient$[]$
28: **end for**
---

| Method | error | log(error) |
|---|---|---|
| stochastic | 0.0956 | -2.3476 |
| baseline non-stochastic | 0.0952 | -2.3515 |
| FISTA | 0.1662 | -1.7946 |
| momentum+stop on gradient sign change | 0.0699 | -2.6607 |
| momentum+stop on gradient sign change + $\eta = 1$ | 0.0701 | -2.6578 |
| momentum+stop on gradient sign change + $\eta = 1$ at 4x rate | 0.0751 | -2.5889 |
| auto-adjusted rate for momentum+stop on gradient sign change | 0.0631 | -2.7630 |
| auto-adjusted rate for momentum+stop on gradient sign change + $\eta = 1$ | 0.0654 | -2.7272 |

Table 1: Performance of algorithms on handwritten dataset

rate becomes rough but still converges. The non-stochastic method is able to make the passes faster, because it performs the same accumulation of partial gradients, but saves the overhead of updating the weights after each training case (or batch). However even adjusted for time, the stochastic method performs faster. It is possible to compute the non-stochastic gradient in parallel by multiple threads but we have not implemented this. We used this example of the non-stochastic descent as a baseline for the FISTA-based momentum methods.

The summary of training errors in the momentum methods can be found in Figure 7A, and the mean square errors after 10,000 training passes are listed in the Table 1.

The unmodified FISTA algorithm with $\lambda = 0$ performed on this task out of its domain worse than the non-momentum baseline. Adding the momentum stop in the dimenstions with gradient sign change produced a substantial improvement over the baseline. Fixing the parameter $\eta = 1$ produced a close result to not fixing $\eta$, but with a less rough curve. We tested if the smoothness indicated that setting

$\eta = 1$ could accommodate a substantial increase in descent rate by re-running the algorithm at a 4x rate (0.2 instead of 0.05), and while this run did not diverge, we did observe a higher error rate.

Finally, we attempted to devise an algorithm that acts similar to the TFOCS dynamic evaluation of $L$ but using the ratios of mean square values of gradient dimensions that change or not change sign as an indication of roughness. The rapid growth of gradient dimensions after sign change is seen as a beginning of a divergence, that causes the reduction of descent rate. This algorithm allowed training at a substantially higher rate in the first few thousands of passes but then flattened out. The automatically determined rate is close to the empirically found 0.05, and is higher in the initial passes where it reaches higher values, but then drops to the lower values (Fig. 7B). It is possible that the chosen criteria were not aggressive enough, and can be improved.

While event with the momentum descent the stochastic methods specialized for NN training can still achieve faster speeds (Fig. 7C-D), we have demonstrated that our more general optimization still represent a major improvement over both simple gradient descent and plain FISTA in a different domain.

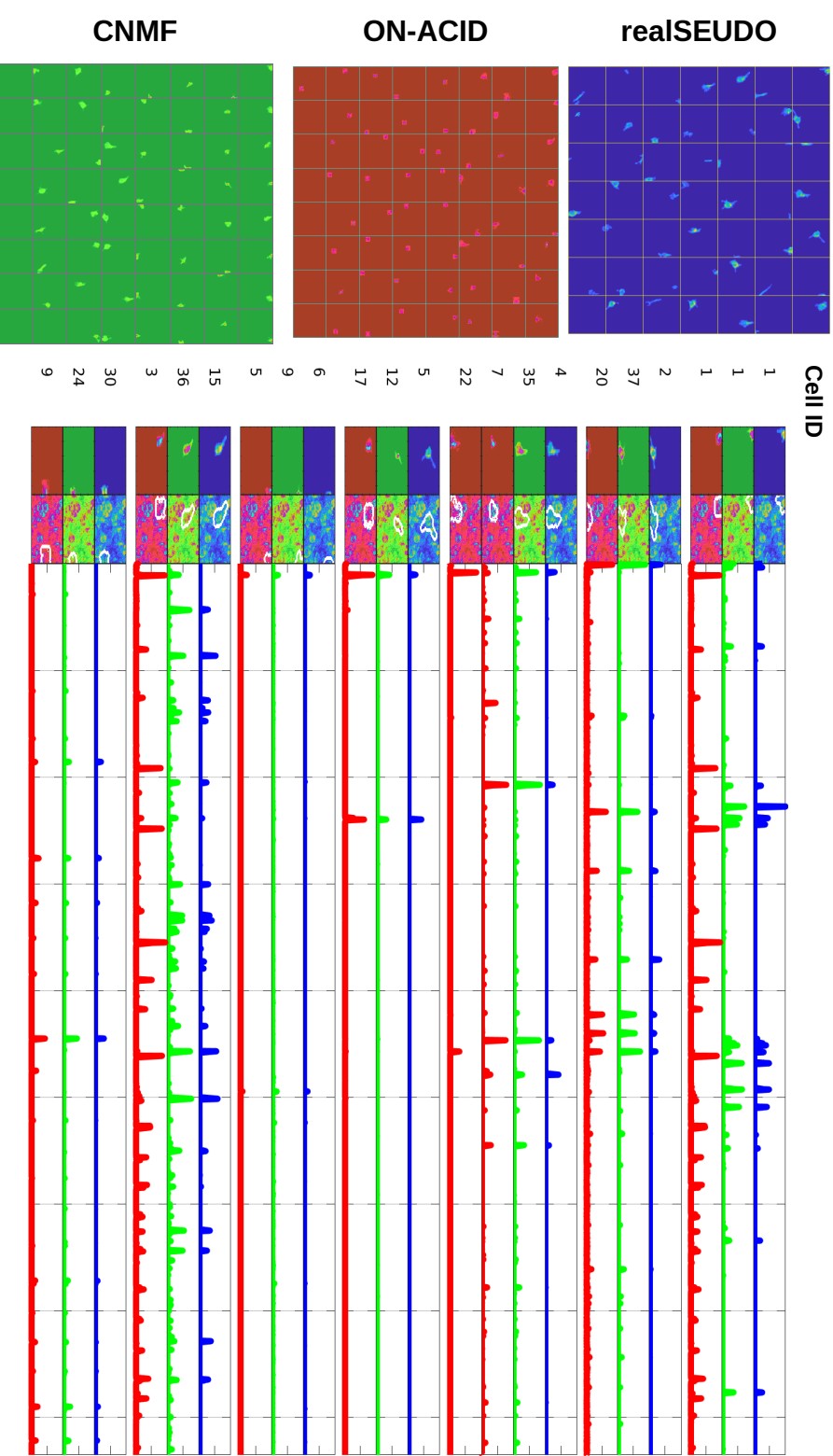

Figure 4: The whole set of cells detected by realSEUDO (blue), OnACID (red), and CNMF (green) in one movie, with selected time traces of matching cells.

# realSEUDO output:

**Cell ID**

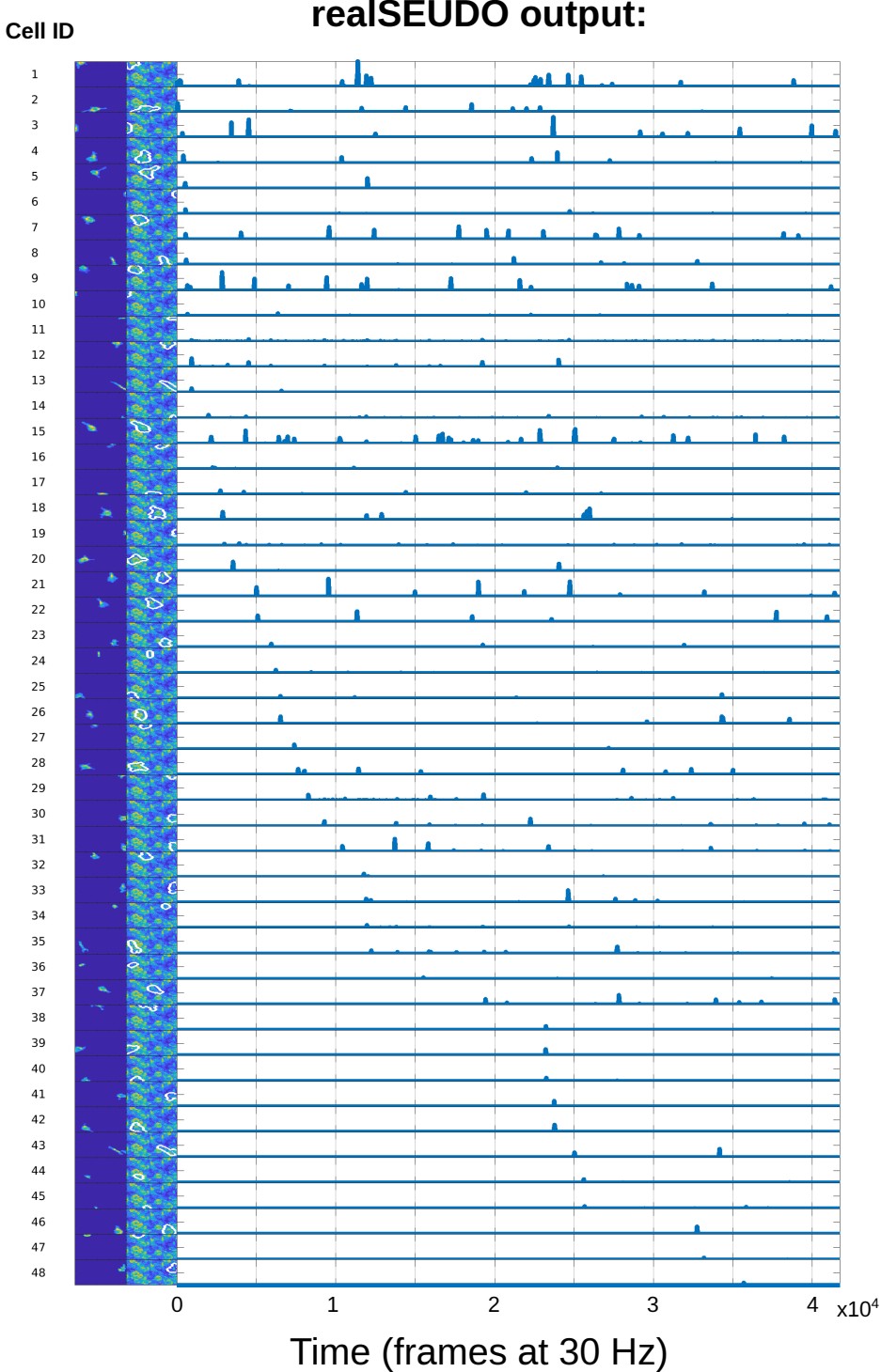

Time (frames at 30 Hz)

Figure 5: Example realSEUDO cells and traces from a single patch, ordered by the discovery time.

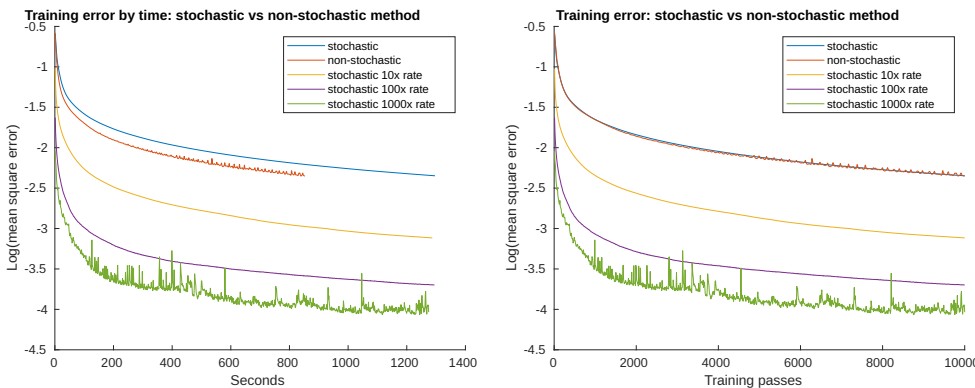

Figure 6: Comparison of different algorithm's learning curves on handwritten datasets. Left: mean-squared error (MSE) as a function of optimization time. Right: MSE as a function of training passes.

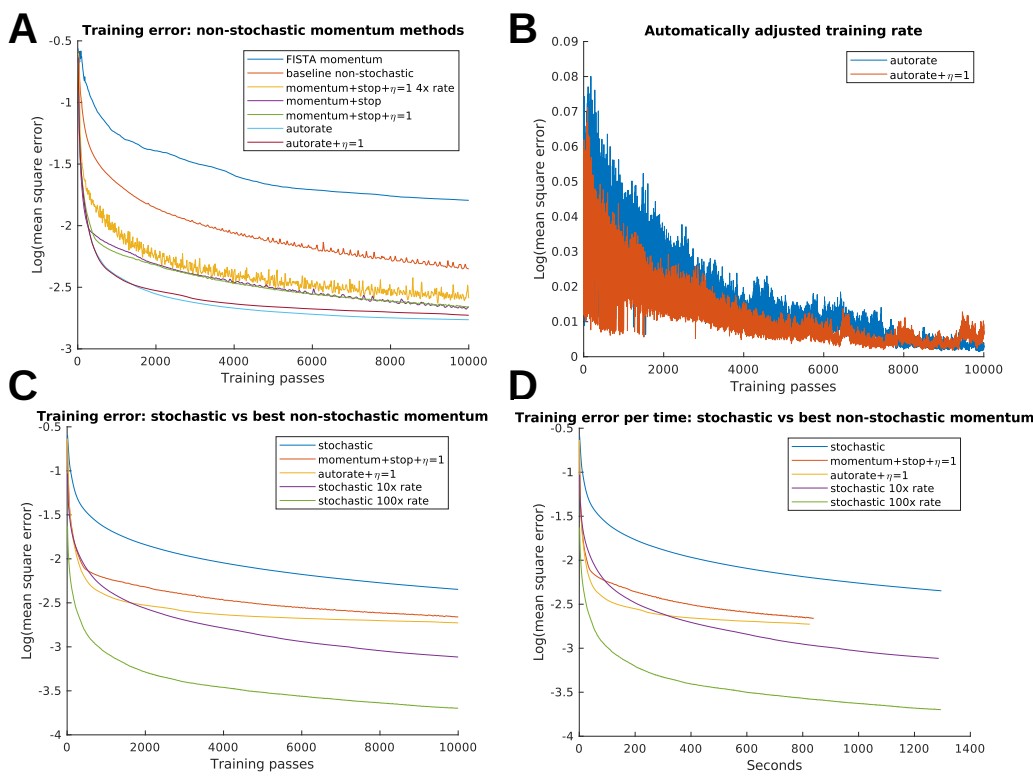

Figure 7: Comaprison of training curves for different algorithms. A: Training MSE as a function of training passes for different variants of the improved FISTA algorithm. B: Training MSE improvement when setting $\eta = 1$. C: Training MSE as a function of training passes for the best tested non-stochastic methods vs. momentum-improved FISTA D: Training MSE as a function of optimization time for the best tested non-stochastic methods vs. momentum-improved FISTA

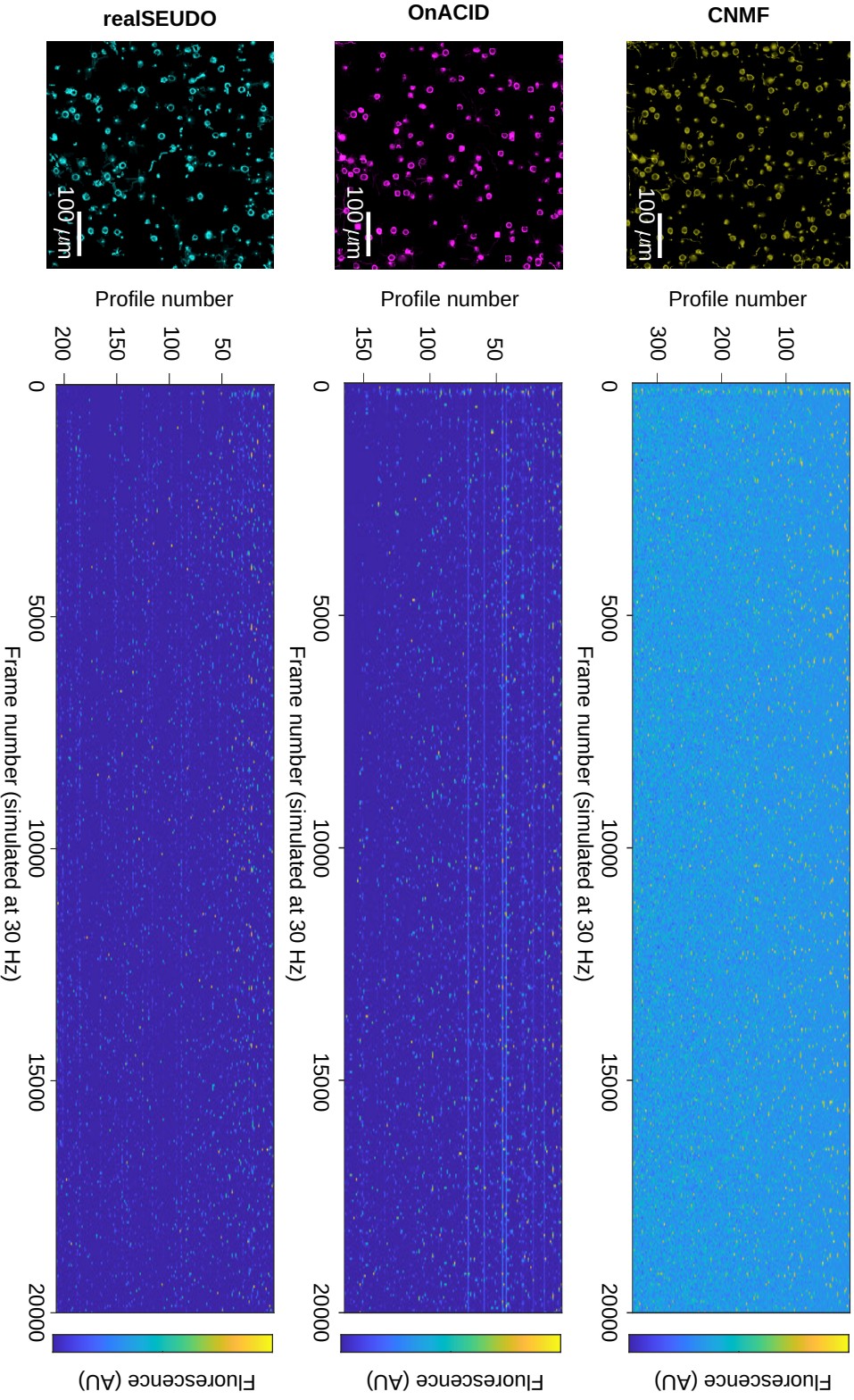

Figure 8: Full sets of strongly-paired traces from CNMF, OnACID and realSEUDO.

