# OpenReview forum: "realSEUDO for real-time calcium imaging analysis"
_NeurIPS.cc/2024/Conference — NeurIPS 2024 poster_

### Official Review · Reviewer_2uGa · 2024-06-17

**Soundness:** 4
**Presentation:** 3
**Contribution:** 3
**Rating:** 7
**Confidence:** 5

**Summary:**

The authors extend an available algorithm called SEUDO for extracting time traces from two-photon imaging data robustly to the real-time close-loop setting. The algorithm can simultaneously extract ROIs and time traces. The algorithm performs on par with state-of-the-art offline algorithms and better than state-of-the-art online algorithms.

**Strengths:**

* The paper is well written and the motivation is clear and the summary of the literature concise and compelling.
* While the algorithmic components of the work are not new in themselves, they are combined in a convincing way to solve the problem laid out.
* The speed of the algorithm is impressive, at satisfying performance compared to the SOTA.

**Weaknesses:**

* The main improvement come from engineering the implementation of SEUDO such as to run in close to real time (is that a weakness or strength?).
* Lines 228: The merging procedure seems quite ad-hoc and heuristic. The logic is only laid out at the end (lines 247), maybe it would help to discuss this earlier.
* Only two algorithms are chosen for comparison.
* Fig. 2B is very hard to read and understand. Does this mean, OnACID produces no false alarms and no ambiguous units?

**Questions:**

* A lot of engineering work has gone into the fast implementation of the online algorithm using C++ and various bells and whistles. Have the authors considered a GPU based implementation e.g. in jax and how would that compare in performance?
* Where does equation 7 come from? Min operations are quite sensitive to outliers. The details won’t matter, but in the spike sorting literature, a threshold of n*0.6745(median (abs(x))) is often used for detecting spikes, which is indeed a multiple of a robust estimate of the noise sd without the spikes.
* Why was the tool not compared to the other online algorithm FIOLA?
* Why was the plain CNMF algorithm used and not one of the more advanced implementations/methods or an additional algorithm?
* Doesn’t Fig. 2C speak very clearly for the results of the OnACID algorithm?
* Lines 348: Why do the authors say that a c++ implementation makes the code open source?

**Limitations:**

Adequately adressed.

---

> ### Author Rebuttal · Authors · 2024-08-06
>
> **Weaknesses:**
> 1. **The main improvement come from engineering the implementation of SEUDO such as to run in close to real time (is that a weakness or strength?).**
>
> **Our response:** Thank you, and we believe that our improvements represent a strength of our approach as we are able to successfully improve and implement a previously slow and unyieldly algorithm in real-time.
>
> 2. **Lines 228: The merging procedure seems quite ad-hoc and heuristic. The logic is only laid out at the end (lines 247), maybe it would help to discuss this earlier.**
>
> **Our response:** We appreciate this suggestion and agree that this section can be improved. In the revision we will begin this section with the logic of our merging procedure to clarify the steps we take.
>
> 3. **Only two algorithms are chosen for comparison.**
>
> **Our response:** In our work we selected the most relevant algorithms. As a baseline we selected CNMF as a standard, SOTA, well-maintained method representative of off-line methods. Comparing to other off-line methods we did not believe to be relevant given that we expect them all to perform at or below the level of CNMF (see, e.g., Song et al. 2021 and Charles et al. 2022 for comparisons between offline methods). OnACID currently represented the only CPU-based online method in existence and so our comparisons had to be to it.
>
> 4. **Fig. 2B is very hard to read and understand. Does this mean, OnACID produces no false alarms and no ambiguous units?**
>
> **Our response:** Yes, that is correct. In the NAOMi data, under the definition from Song et al. 2021 of a found cell (spatial overlap of over 50% and temporal correlation >0.5) OnACID performed very conservatively in the best parameter set we could find for it. This meant that it actually reduced quite a bit of noise as well as true transients. Thus the correlations are overall higher, however the total number of cells was reduced. Additionally, OnACID is still very slow (<10 Hz) meaning that it is not truly "real-time" for this data.
>
> **Questions:**
>
> 1. **A lot of engineering work has gone into the fast implementation of the online algorithm using C++ and various bells and whistles. Have the authors considered a GPU based implementation e.g. in jax and how would that compare in performance?**
>
> **Our response:** We had considered both GPU and FPGAs as alternative approaches that could have been viable. While potentially beneficial in some ways, our ultimate goal was integration of our online cell-finding into a larger closed-loop pipeline. In this case bringing the pipeline closer to the imaging is necessary, and most imaging groups don’t have high performance GPUs or the expertise to install and make use of FPGAs. We thus decided that a purely CPU approach would reduce the hardware complexity in the long term as we work with experimental groups to implement this method in closed loop systems.
>
> 2. **Where does equation 7 come from? Min operations are quite sensitive to outliers. The details won’t matter, but in the spike sorting literature, a threshold of n*0.6745(median (abs(x))) is often used for detecting spikes, which is indeed a multiple of a robust estimate of the noise sd without the spikes.**
>
> **Our response:** Equation (7) simply comes from the desire to assess the width of the pixel histogram mode (the lower half of the histogram as there is usually an asymmetric tail due to active pixels). It is true that the min is usually sensitive to noise, however we actually compute this min on the denoised data which provides a more stable minimum value. We agree that this is unclear in the paper and will clarify this detail in the text.
>
> 3. **Why was the tool not compared to the other online algorithm FIOLA?**
>
> **Our response:** FIOLA is an online algorithm for demixing, and not cell identification. Specifically it is more of an optimized pipeline that relies on pre-computed ROIs to extract time-traces from online. The authors mention this in the FIOLA paper (page 1420). Thus we did not feel it is an appropriate comparison point. In fact we can envision parts of the realSEUDO concept being integrated into their very excellently engineered pipeline that solves other important problems, such as motion correction and spike extraction.
>
> 4. **Why was the plain CNMF algorithm used and not one of the more advanced implementations/methods or an additional algorithm?**
>
> **Our response:** We specified CNMF, however used the most recent version available in the CaImAn package provided by the Simons institute. We felt that it represented one of the state-of-the-art methods and was related to OnACID which gave a good complementary point of reference of what the same algorithm could look like on-line and off-line.
>
> 5. **Doesn’t Fig. 2C speak very clearly for the results of the OnACID algorithm?**
>
> **Our response:** OnACID had varying results, performing better in the NAOMi simulated data, for example, than the CA1 real dataset. Specifically we found that OnACID performed quite well in the simulated NOAMi data under conservative parameters. This is primarily due to the high thresholding value that both reduced noise in the estimated time-traces. However this higher threshold also removed smaller transients (see e.g., Profile 2 in Fig. 2D). Moreover OnACID was considerably slower and unable to run at real-time speeds (<10 Hz as opposed to realSEUDO's >65 Hz, see Fig. 2C).
>
> 6. **Lines 348: Why do the authors say that a c++ implementation makes the code open source?**
>
> **Our response:** We simply wanted to make sure the community understood that the code was not written completely in MATLAB, which is not technically open source even when released. The C++ code has a wrapper in MATLAB, and we are working on similar wrappers for Python. We will clarify this point in the paper.

---

> > ### Comment · Reviewer_2uGa · 2024-08-08
> >
> > Thanks for the clarification. I maintain my score, and would vote clearly in favor of "accept".
> >
> > Regarding the use of min: Consider using the 5th percentile or similar.

---

> > > ### Author Response · Authors · 2024-08-08
> > > **Thank you**
> > >
> > > Dear reviewer,
> > > Thank you and I am glad that we were able to clarify! Your point about the 5th percentile is well taken and we will definitely test that out.

---

### Official Review · Reviewer_uJzY · 2024-07-11

**Soundness:** 3
**Presentation:** 3
**Contribution:** 2
**Rating:** 6
**Confidence:** 3

**Summary:**

In this study, the authors created a real-time version of the “Sparse Emulation of Unused Dictionary Objects” (SEUDO) algorithm termed realSEUDO, that can process multi-photon calcium imaging (CI) data in real time. The goal of this approach is to overcome the limitations of batch processing methods commonly employed in CI analysis by processing brain activity data in real-time, thus enabling closed-loop neuroscience investigations. The authors identified several places that the original SEUDO algorithm warranted improvements, implemented the changes and compared their method to other current online methods, with both simulated and real datasets.

**Strengths:**

This research presents a novel real-time CI data analysis method that is important for closed-loop investigations.  The realSEUDO algorithm as presented here, incorporates a significant number of optimizations over the original SEUDO, including a faster C++ implementation, parallel processing, and an efficient feedback loop for real-time cell identification that improves the computational load. In this way they can achieve much faster performance compared to other online methods (OnACID). Another important strength is its ability to match sources across non-overlapping parches, which reduces the computational load. The authors use both simulated and real datasets to evaluate the performance of their algorithm. This method could also become very useful in online-analysis of voltage traces that higher speeds can have even more benefits.

**Weaknesses:**

While the performance with a small number of neurons is impressive there are two drawbacks.
First, the testing datasets only include a very limited number of neurons and the scalability with much larger datasets are not presented. For example, current methods such as with a mesoscope  (Sofroniew et al. 2016), allow for much larger recording samples in the of thousands of neurons. From the figure 3c we can see that the algorithm drops in performance quite fast and when much larger samples are included, it could potentially perform worse than existing methods, something that remains to be tested.
Lastly, calcium sensors already limit our ability to extract with fine temporal precision the neural activity thus the usefulness of a method that far exceeds that temporal resolution remains limited.
Another potential issue is to validate the model with more typical cortical recordings that have higher level of background fluorescence, and how it deals with overlapping sources of fluorescent signal.

**Questions:**

By facilitating the real-time processing of CI data, the paper shows the potential to improve the accuracy and efficiency of closed-loop studies. The claims of broad applicability would be strengthened by more extensive testing under various experimental situations and with diverse kinds of brain data. Additionally, it would be great if they could characterize and show the performance with different patch overlaps. Finally, while the paper discusses the integration with motion correction algorithms from OnACID, integration of all preprocessing steps such as motion correction within the realSEUDO pipeline would further widen its potential usability.

**Limitations:**

The authors have a limitation section that addresses potential issues with their algorithm.

---

> ### Author Rebuttal · Authors · 2024-08-06
>
> **Weaknesses: While the performance with a small number of neurons is impressive there are two drawbacks. First, the testing datasets only include a very limited number of neurons and the scalability with much larger datasets are not presented. For example, current methods such as with a mesoscope (Sofroniew et al. 2016), allow for much larger recording samples in the of thousands of neurons. From the figure 3c we can see that the algorithm drops in performance quite fast and when much larger samples are included, it could potentially perform worse than existing methods, something that remains to be tested. Lastly, calcium sensors already limit our ability to extract with fine temporal precision the neural activity thus the usefulness of a method that far exceeds that temporal resolution remains limited. Another potential issue is to validate the model with more typical cortical recordings that have higher level of background fluorescence, and how it deals with overlapping sources of fluorescent signal.**
>
> **Our response:** We thank the reviewer for noticing the positive aspects of our work and the novelty of our contribution. We would like to address the three main weaknesses that concern this reviewer: 1) scaling to very large imaging, 2) speed of calcium sensors and the need for even faster processing, and 3) the potential for overlapping sources of fluorescence.
> 1. We agree that newer methods have been developed that aim to improve the number of sources that can be detected simultaneously. It is also the case that 500um-by-500um fields of view are still the standard for most labs and our patching approach easily extends to these sizes. We of course are interested in testing the boundaries of the capabilities of our method, and we will mention this limitation presented by the reviewer in the discussion and limitations section.
> 2. The reviewer is right in mentioning the speeds of most calcium sensors. However we believe that two main points justify even faster processing. For one, new indicators such as GCaMP8f are even faster and may require more than 30Hz imaging to properly capture (and thus even faster processing). Second, the cell finding step is only one part of a closed loop system. The idea is to enable on-line experimental perturbations, which will also require online model fitting and the computation of the right perturbation (e.g., stimulation or experimental change) after each cell’s activity is demixed by realSEUDO. Thus if realSEUDO takes even less time, there is more time for later analysis steps that will enable true closed-loop experiments
> 3. In terms of overlapping sources of fluorescence, the reviewer is correct that this is a critical issue, especially in on-line demixing where sources of fluorescence are unknown. Thus, a key design choice in selecting SEUDO as the core of our approach was its ability to be robust to overlapping sources of fluorescence even when one source was unknown. The original SEUDO paper, Gauthier et al. 2022, demonstrated this ability, which we further demonstrate remains present in realSEUDO by counting the number of ``false transients’’ (i.e., bursts of activity due to bleedthrough from other sources rather than the cell itself) in Section 4, bottom of Page 7. Specifically, realSEUDO can continue to remove false bleedthrough activity despite the many overlapping sources of fluorescence in the CA1 tissue that we tested realSEUDO on.
>
> **Questions: By facilitating the real-time processing of CI data, the paper shows the potential to improve the accuracy and efficiency of closed-loop studies. The claims of broad applicability would be strengthened by more extensive testing under various experimental situations and with diverse kinds of brain data. Additionally, it would be great if they could characterize and show the performance with different patch overlaps. Finally, while the paper discusses the integration with motion correction algorithms from OnACID, integration of all preprocessing steps such as motion correction within the realSEUDO pipeline would further widen its potential usability.**
>
> **Our response:** Again we appreciate the reviewer’s positive outlook on this work. We aimed to focus our validation on datasets with rigorous ground truth, which included real data from Hippocampus and Visual cortex, along with the realistic simulations. As the Hippocampus data had much richer validation data (i.e., it was previously annotated by the authors of the study not just in terms of cells but also time-traces), we focused on this dataset for our work. We will mention these aspects of our study in more detail in the discussion to clarify. Moreover we will make mention of the need to integrate out advance, which focused on one critical step of the pipeline, with other steps such as motion correction that have been addressed by other researchers.

---

> > ### Comment · Reviewer_uJzY · 2024-08-09
> >
> > I would like to thank the authors for the response to my comments and the clarifications that will appear in the text.

---

### Official Review · Reviewer_Xdei · 2024-07-12

**Soundness:** 3
**Presentation:** 3
**Contribution:** 2
**Rating:** 6
**Confidence:** 2

**Summary:**

The authors present an online method for neuron profile and trace estimation in multi-photon calcium imaging. Their approach enhances the SEUDO algorithm by running it on a frame-by-frame basis and periodically updating profiles using a patching mechanism across space and time. The implementation leverages different degrees of stability, namely temporary and stable. The proposed method is benchmarked against both an online method and an offline method using simulated and real data.

**Strengths:**

- The manuscript provides extensive details about the implementation.
- The impact of the proposed solution is clearly stated and is highly relevant for neuroscientists working with CI acquisitions, addressing several real issues such as empty initialization of profiles and scalability.

**Weaknesses:**

- The work largely relies on previously described algorithms like FISTA and SEUDO. Since a significant portion of the method describes these algorithms in detail, it is unclear what the key contributions of this work are.
- The evaluation was conducted on synthetic data (Naomi) and a real dataset (36 videos). Including additional sources like 2p movies from the Allen Brain Observatory would have enhanced the credibility of the results.
- The given metrics should be improved. While the images are good, providing tables with quantitative metrics can enhance the overall presentation and support the claims made. In addition to correlation, metrics like IoU could be evaluated on neuron profiles.
- The experimental setting lacks an ablation study on the main components of the work, such as automatic cell recognition (e.g. with and w/o X_temp) and patching.

**Questions:**

1. What are the key methodological differences between SEUDO and realSEUDO?
2. How did you choose the benchmark datasets?
3. Why correlation was selected as the main (and only) metric?
4. How do the authors justify the lack of an ablation study on the main components of the work? Can the authors describe the quantitative improvement of their method compared to existing methods they rely on?

**Limitations:**

The authors have adequately addressed the limitations.

---

> ### Author Rebuttal · Authors · 2024-08-06
>
> **Weaknesses:**
> 1. **The work largely relies on previously described algorithms like FISTA and SEUDO. Since a significant portion of the method describes these algorithms in detail, it is unclear what the key contributions of this work are.**
>
> **Our response:** While we leveraged prior developments in the field, we would like to note that the ability to create a method that met the criteria for real-time demixing was not a trivial combination of these tools. Several significant changes, from efficient computation, implementation, and model modification had to work together in conjunction just to have SEUDO run in realtime. Moreover we developed a feedback loop for generating new ROIs (note that SEUDO only identifies time-traces given a-priori known ROIs) in an online manner.
>
> 2. **The evaluation was conducted on synthetic data (Naomi) and a real dataset (36 videos). Including additional sources like 2p movies from the Allen Brain Observatory would have enhanced the credibility of the results.**
>
> **Our response:** While more results are always better, we carefully selected datasets that had reliable ground truth, either through data generation (e.g., the NAOMi simulation) or through rigorous prior assessment (e.g., the data from Gauthier et al. 2022). Other datasets do not have very reliable ground truth and thus would not serve as well to validate our approach. Moreover data from the Allen Brain Observatory is only disseminated as traces and ROIs, not as raw data. This is especially problematic as the SEUDO paper (Gauthier et al. 2022) previously showed that these are not reliable decompositions of the data.
>
> 3. **The given metrics should be improved. While the images are good, providing tables with quantitative metrics can enhance the overall presentation and support the claims made. In addition to correlation, metrics like IoU could be evaluated on neuron profiles.**
>
> **Our response:** We appreciate the suggestion. As the IoU implies that the full set of pixels identified is the important quantity, we instead compute the “unique neurons found”. This is since in functional imaging we instead typically need to know that the time-traces 1) correspond to real neurons in the data and 2) accurately reflect the temporal activity that will be used to study neural activity with respect to stimuli and behavior. The Unique Neurons Found requires both that ROIs well align with known ROIs spatially without the strict pixel counting that might not accurately reflect what a neuroscientist deems “success” in regards to cell finding. We will clarify our criteria for selecting these metrics and how the metrics used reflect these key requirements for validation in the results and discussion. We also selected to place the comparison as bar plots to situate them visually next to the images of the identified components and example time-traces, compartmentalizing that example.
>
> 4. **The experimental setting lacks an ablation study on the main components of the work, such as automatic cell recognition (e.g. with and w/o X_temp) and patching.**
>
> **Our response:** We agree with the importance of ablation studies in algorithmic development, however the overall algorithm is quite simple. In fact directly adding ROIs to the dictionary instead of using X_temp is, as the reviewer saw, really the only section that can be ablated without effectively breaking the ROI selection loop. We have previously had ROIs added directly to the dictionary, which has resulted in a lot of noise being treated as signal. We did directly test an ablation study focused on the speed of computation, slowly adding every improvement. We noted the increasing framerate on the last paragraph before Section 3.2 on Page 5, which shows the necessity of each improvement to the overall performance.
>
> **Questions:**
> 1. **What are the key methodological differences between SEUDO and realSEUDO?**
>
> **Our response:** There are differences in SEUDO and realSEUDO on both the computation and algorithmic levels. On the algorithmic level we introduced a full feedback loop that identifies ROIs. SEUDO, as derived and presented in Gauthier et al. (2022) is a time-trace estimation method only. It relies on pre-defined ROIs and focuses on robust estimation of the activity for each ROI. We add the ability to use the residuals from SEUDO to detect in an online manner new ROIs to iteratively build a dictionary of ROIs from nothing.
>
> 2. **How did you choose the benchmark datasets?**
> We chose the benchmark datasets as datasets with extensive information (either through simulation in NAOMi data or manual expert annotation in the Gauthier et al. dataset) on both the ROIs and also the time-traces (e.g., our false positive comparisons). These properties gave us the best targets to validate our data quantitatively.
>
> 3. **Why correlation was selected as the main (and only) metric?**
>
> **Our response:** Correlation as a metric was also predicated on ROI overlap. Essentially our “Unique cells detected” number combined both temporal correlation and spatial information to claim that we detected a cell well. We isolated the correlations into a separate histogram to highlight the success in identifying the time-traces; a quantity that is vital to the fitting of computational neuroscience models.
>
> 4. **How do the authors justify the lack of an ablation study on the main components of the work? Can the authors describe the quantitative improvement of their method compared to existing methods they rely on?**
>
> In terms of completely removing parts of our algorithm, the main parts are SEUDO and the feedback for specifying ROIs. Removing either would break the loop and not allow for the algorithm to be run at all. In terms of speed, we do show how each change we made improves speed (a type of ablation) showing that all different improvements we made are necessary to achieve the full speed-up. These numbers are detailed on the last paragraph before Section 3.2 on Page 5.

---

> > ### Comment · Reviewer_Xdei · 2024-08-09
> >
> > I thank the authors for their clarification. Regarding the evaluation, I acknowledge the authors' efforts in choosing the right datasets. However, I would find the results more convincing if another source of real data were used. In this regard, the Allen Visual Coding - 2P dataset, in addition to extracted traces and ROIs, also provides raw movies on AWS.
> >
> > Aside from this, I also acknowledge that all technical aspects were well addressed, and I find the authors' motivation for not using segmentation metrics convincing. I will update my original assessment to reflect my new perspective.

---

> > > ### Author Response · Authors · 2024-08-12
> > > **Thank you**
> > >
> > > Thank you for pointing us to the availability of the raw Allen Visual Coding - 2P dataset on AWS. We will certainly be making use of this resource as we continue to assess and develop new methods for 2P data processing. We are also thankful that our clarifications were helpful in understanding our choices in metrics and will make sure these points are clear in the final manuscript.

---

### Official Review · Reviewer_86dC · 2024-07-22

**Soundness:** 4
**Presentation:** 4
**Contribution:** 3
**Rating:** 8
**Confidence:** 4

**Summary:**

The paper presents a novel online method for cell detection and fluorescence estimation from streaming calcium imaging (CI) data, called realSEUDO. Building upon the SEUDO algorithm, realSEUDO introduces significant advancements in real-time cell identification and fluorescence estimation. The authors effectively address the challenge of high-speed processing by leveraging parallel computing and profile matching techniques across patches and temporal dimensions.

**Strengths:**

The paper introduces realSEUDO, a groundbreaking online method for real-time cell detection and fluorescence estimation in calcium imaging data. It significantly outperforms existing methods like OnACID and CNMF in both processing speed and accuracy, achieving frame rates of 80-200 fps while maintaining high detection precision. The innovative integration of temporal and spatial profile matching, combined with efficient patch-based parallel processing, marks a substantial advancement in the field. Detailed experimental validation further underscores realSEUDO’s superior performance, adaptability, and potential for scaling with faster imaging technologies.

**Weaknesses:**

While the paper presents a novel and efficient approach with realSEUDO, several limitations are apparent. The reliance on specific hardware configurations, such as Linux CPU performance modes, could limit accessibility and generalizability. Additionally, the implementation primarily in C++ with MATLAB wrappers, while effective, could benefit from broader integration into Python pipelines to enhance usability. The paper also assumes access to external motion correction algorithms and does not address the full pipeline holistically, which may impact its applicability in scenarios without these additional tools.

**Questions:**

N/A

**Limitations:**

The paper presents notable advancements with realSEUDO but has some limitations. Firstly, the performance of the system is highly dependent on specific hardware configurations and the proper setting of Linux CPU manager modes, which may not be universally applicable. Additionally, the reliance on MATLAB for prototyping, combined with limited Python integration, may restrict its accessibility and ease of use for a broader audience. Moreover, the paper focuses on cell detection without fully integrating or addressing the motion correction and complete processing pipeline, which could limit its applicability in real-world scenarios where such components are not readily available.

---

> ### Author Rebuttal · Authors · 2024-08-06
>
> **Weaknesses: While the paper presents a novel and efficient approach with realSEUDO, several limitations are apparent. The reliance on specific hardware configurations, such as Linux CPU performance modes, could limit accessibility and generalizability. Additionally, the implementation primarily in C++ with MATLAB wrappers, while effective, could benefit from broader integration into Python pipelines to enhance usability. The paper also assumes access to external motion correction algorithms and does not address the full pipeline holistically, which may impact its applicability in scenarios without these additional tools.**
>
> **Our response:** We appreciate the reviewer’s positive comments and feedback. We agree that the additional limitations of the current infrastructure should be more explicit. Some drawbacks, such as additional wrappers in Python (as well as pure C++ integration into imaging pipelines) are a primary focus of future development in this project. We are also testing the exact limitations of the hardware impact on performance. We do believe that in the long-run CPU integration will allow for more generalizability than, e.g., GPU or FPGA approaches, and will add more text in the discussion to discuss the pros and cons of each. Finally, as online motion correction was well addressed by others (e.g., in the OnACID work) we instead focused on the less-considered cell finding. It is of course important, as the reviewer indicates, that full pipelines work well together and so integration of prior methods are a primary consideration as incorporate these pipelines into full closed-loop systems.
>
> **Limitations: The paper presents notable advancements with realSEUDO but has some limitations. Firstly, the performance of the system is highly dependent on specific hardware configurations and the proper setting of Linux CPU manager modes, which may not be universally applicable. Additionally, the reliance on MATLAB for prototyping, combined with limited Python integration, may restrict its accessibility and ease of use for a broader audience. Moreover, the paper focuses on cell detection without fully integrating or addressing the motion correction and complete processing pipeline, which could limit its applicability in real-world scenarios where such components are not readily available.**
>
> **Our response:** We thank the reviewer for noting our advancements and seeing the potential in our work. As the reviewer mentioned, the primary goal of this work is to address a critical step in the pipeline for real-time cell detection. We agree that with every advance there will be limitations and that these limitations should be clearly delineated. Moreover we agree that the stage of the pipeline we address does not live in a vacuum and should be mentioned in the context of the other stages of the pipeline to best inform readers of when/how they can use the method. We will thus discuss in more detail the limitations raised by the reviewer and the steps we are actively taking to expand the ease of others to adopt and implement realSEUDO. These include additional integration tools into Python on the user’s end, as well as building in prior motion correction algorithms that will be a part of the complete pipeline development.

---

### Author Rebuttal · Authors · 2024-08-06

We thank the reviewers for their time and effort in reviewing our manuscript. We are encouraged that the reviewers recognized the relevance, capabilities, and potential impact of our work. We take the time here to respond to the primary concerns the reviewers raised, with more detailed responses to each reviewer.
1. **Novelty of approach:** Reviewers were mixed in their assessment of novelty of the approach. We would like to take the time to clarify that while we did leverage prior developments in the field, specifically FISTA and SEUDO, we significantly improved these algorithms at multiple levels to create a new method that met the criteria for real-time demixing. In addition, none of these methods has a cell-finding feature, and so the individual improvements (efficient computation, implementation, and model modification), the combination, and the creation of the feedback loop to turn fast time-trace estimation into fast cell finding we believe was not a trivial combination of these tools.
2. **Integration into more general pipelines:** Reviewers 86dC and uJzY asked about further integration into current pipelines (e.g., including the motion correction and having additional wrappers). We agree, and are working on complete integration. As online motion correction already exists, this work focuses on a different critical piece of the real-time analysis puzzle: cell detection. While some engineering work will be required to connect these pieces, we felt that the creation of a non-existent piece should be the main focus of this paper. Similarly, we created a wrapper for our C++ code in MATLAB for easy testing, and will be developing similar Python scripts (or even moving the whole pipeline into C++) to best reach a wider audience. We will discuss these additional pieces and the future steps to integrate realSEUDO into full pipelines in the limitations and discussion sections.
3. **Choice of datasets:** Reviewer Xdei wondered about our choice in validation data and if this presented a limitation in our work. Our primary goal in dataset selection was to choose datasets that can validate the estimated time-traces which are vital for to model neural activity and relate it to behavior. Specifically, time-traces are also the most prone to errors in real-time cell finding as unknown cells can bleed into known cells, creating false activity. Both the NAOMi simulation (Song et al. 2021) and the SEUDO dataset  (Gauthier et al. 2022) have both spatial and temporal annotation. NAOMi provides visual-cortex like data with ground truth traces and the SEUDO data provides dense CA1 data with true/false annotations for individual activity bursts. We further analyzed the third dataset included in the OnACID package, for which we could only assess spatial ROIs.
4. **Quantification and metrics:** Reviewer Xdei mentioned other potential metrics, such as the intersection-over-union metric. We agree with the reviewer that assessing spatial matches is important and we wish to clarify that while we did not use the IoU to validate our approach, we did use spatial information in our “unique cells detected” (Fig. 2B) which considers both spatial overlap and temporal correlations of the time-traces to identify if a found component is a true positive. We find that integrating the spatial and temporal information together into a hit is overall more informative in functional fluorescence microscopy as the exact pixel count is less important than cell ID (i.e., overall location) and time-trace accuracy.
5. **Choice of algorithm comparisons:** Reviewer 2uGa wonders about the selection of comparison algorithms. There are very few comparison points for online processing of calcium imaging data. OnACID represents the primary method available to the community. Other methods, such as FIOLA are more of a pipeline and actually rely on pre-computed ROIs for fast time-trace estimation. We thus compared to OnACID, which while accurate at times can be quite slow. We also compared realSEUDO to a state-of-the-art off-line method CNMF (using the latest version available through Simons) to show how close (or far) on-line methods are as compared to off-line methods. Many off-line methods exist, and other works compare off-line methods (e.g., Song et al. 2021 and Charles et al. 2022) which informed our decision of selecting one of the consistently highest performing off-line algorithms.

---

### Decision · Program_Chairs · 2024-09-25

**Decision:**

Accept (poster)

**Comment:**

This paper extends a previously developed algorithm called SEUDO for extracting time traces from two-photon imaging data robustly also in challenging real-time close-loop setting. The algorithm can simultaneously extract ROIs and time traces. The algorithm performs competitively with state-of-the-art offline algorithms and better than state-of-the-art online algorithms. The results and concepts are well explained, broadly accessible and potentially very impactful for closed-loop neuroscience experiments.